# Are Perceptions of Government Intervention Related to Support for Prevention? An Australian Survey Study

**DOI:** 10.3390/healthcare11091246

**Published:** 2023-04-27

**Authors:** Anne Carolyn Grunseit, Eloise Howse, Julie Williams, Adrian Ernest Bauman

**Affiliations:** 1School of Public Health, University of Technology Sydney, 15 Broadway, Ultimo, NSW 2007, Australia; 2Prevention Research Collaboration, Charles Perkins Centre, Sydney School of Public Health, Level 6, University of Sydney, Camperdown, NSW 2006, Australia; 3The Australian Prevention Partnership Centre, The Sax Institute, Level 3, 30C Wentworth St, Glebe, NSW 2037, Australia; 4Public Health Services, Tasmanian Department of Health, 2/25 Argyle Street, Hobart, TAS 7000, Australia

**Keywords:** Australia, public opinion, attitudes, prevention, non-communicable diseases, policy

## Abstract

Background: In Australia, despite the success of tobacco control policy interventions, policymakers remain resistant to policy-based approaches to diet, alcohol, physical inactivity and obesity, concerned about community perceptions of such interventions as “nanny-statist”. We examined how people’s general positions on government intervention related to their positions on different preventive policy options. Methods: Data were from a 2018 nationally representative cross-sectional telephone survey of 2601 Australian adults. Survey questions related to endorsement of different conceptualisations of government intervention (nanny state, paternalistic, shared responsibility and communitarian) and support for specific health interventions, using forced-choice questions about preferences for individual/treatment measures versus population/preventive health measures. We analysed associations between scores on different conceptualisations of government intervention and support of different policy options for tobacco and diet, and preferences for prevention over treatment. Results: The Nanny State Scale showed an inverse relationship with support for tobacco- and diet-related interventions, and alternative conceptualisations (paternalistic, shared responsibility and communitarian) showed a positive relationship. Effect sizes in all cases were small. Those aged 55+ demonstrated greater support for policy action on tobacco and diet, and greater preference for systemic rather than individual-level interventions. Conclusion: General disposition towards government intervention, although correlated with support for specific policy actions, is not deterministic.

## 1. Introduction

Non-communicable diseases (NCDs) such as Type 2 diabetes, cancer and cardiovascular disease are chronic, costly conditions that are increasing worldwide and represented 72% of deaths globally in 2016 [1]. In Australia, 38% of the burden of disease could be prevented by addressing key risk factors including high body mass (overweight and obesity), poor dietary patterns and tobacco use [2]. In recent years, there has been greater attention paid to the upstream factors affecting NCD risk factors, with prevention being linked to sustainable development goals [3] and greater emphasis placed on interventions which aim to change the wider systems within which individual behaviours take place [4]. However, the peer-reviewed literature examining NCD interventions seems to still be dominated by individual behaviour measures, with policy and environmental action lagging behind by comparison [4].

In areas such as tobacco control, governmental policy actions in terms of taxation and regulation have succeeded in reducing smoking-related behaviours in Australia [5]. Public support for tobacco control measures has steadily increased, even amongst smokers [6]. Despite this success, there remains some resistance to similar policy-based approaches in areas such as diet, alcohol, physical inactivity and obesity. Current evidence suggests that people tend to support actions that are more education- or information-based [7]; conversely, there is less support for what are termed more “intrusive” interventions [8]. Different actors, including within industry [9] and the media [10], often frame government intervention negatively as indicating a “nanny state” and emphasize individual responsibility for “healthy lifestyles”, calling for better education and information rather than regulation. The concern of public health advocates is that such discourses can affect public opinion and community support for prevention policies [11] as well as discouraging policy action among policymakers [12]. 

Some theorists suggest that social and cultural factors can influence people’s attitudes and perceptions about risk, which in turn shape individual values and worldviews [13,14,15]. Therefore, support for preventive health policies and government intervention could be partly dependent on individuals’ expressed worldviews and ideological beliefs about who is responsible for people’s health [16]. However, most national studies of Australians’ perceptions relevant to the prevention of NCDs have focused specifically on measuring the prevalence of support for obesity prevention [17,18] and sugary drink regulation [19]. The Australian Perceptions Of Prevention Survey (AUSPOPS) is one of the few Australian national-level data collection projects which monitors general attitudes to government intervention for the prevention of NCDs [20]. Various analyses of qualitative and quantitative data from the AUSPOPS study have found limited support for the nanny state conceptualisation of government intervention being widely held, and identified several alternative conceptualisations of such intervention as a “canny investment” as well as showing leadership or being a partner in better health [20]. AUSPOPS data have shown majority and increasing support for a role for government intervention in prevention [21], regardless of age or gender [22]. 

However, how general attitudes towards government intervention for prevention relate to expressed support for specific health policy actions is yet to be tested. The current exploratory study aimed to examine how general positions on government intervention relate to positions across policy options with varying mechanisms and target behaviours/risk factors using the 2018 AUSPOPS survey data. The findings will assist policymakers to understand how to frame messaging for NCD prevention to reach people with different worldviews, and whether those worldviews are related to support for specific government interventions.

## 2. Methods

### 2.1. Survey Design

The AUSPOPS study is a general population survey, first undertaken in June–July 2016, to understand Australian community perceptions of government interventions aimed at reducing lifestyle-related chronic disease [20]. The survey was conducted again in October–November 2018 with the addition of a small number of questions. The analysis reported here is based on the second survey only. Ethics approval was obtained from the University of Sydney Human Research Ethics Committee, approval #2016/141. Informed consent was obtained from all subjects involved in the study.

### 2.2. Sampling

Recruitment was carried out through a commercial sample provider using random digit dialling covering both landline and mobile phone users [23]. Geographic area code data were used to generate a stratified (state by region, capital city/non-capital city) landline sample; however, there was no geographic information available for the mobile phone population. Respondent selection included the person in the household aged 18 years or older whose birthday was next for the landline sample, whereas the person who answered the phone was asked to participate for the mobile sample. To take into account the growing mobile-only population, we used a ratio of 70:30 mobile to landline [24].

### 2.3. Questionnaire

Questions covered health spending priorities, responsibility for health prevention among government and non-government actors, support for specific government health initiatives, self-rated health status and demographic information. A copy of the 2018 questionnaire, along with the corresponding question numbers of the data items described below and in the results, is provided in Appendix A.

### 2.4. Data Items and Measures

#### 2.4.1. Exposure Variable

− Conceptualisations of government intervention

A series of eight questions investigated how the respondent viewed government intervention for prevention in general. Four of the questions were from the original 2016 questionnaire, characterising government interventions as interfering, [25] paternalistic [26] or utilitarian [27] (see Question E5, Appendix A). A further four questions developed from a re-analysis of qualitative focus groups described previously [20] were added to the questionnaire in 2018, which included conceptualisations of government intervention as shared responsibility, supportive of individual agency, explicitly nanny state and futile in the face of personal choices. Response categories ranged from “strongly disagree” to “strongly agree” on a 5-point Likert scale.

#### 2.4.2. Outcome Variables

− Specific policy measures

Respondents were also asked for their views about 13 specific existing or hypothetical policy measures to help Australians be healthy. Respondents were asked “For each of the following government initiatives, please tell me whether you think it shows the government going too far, not far enough or having about the right amount of involvement in helping people be healthy?” (E2, Appendix A). Responses included “going too far”, “about the right amount” or “not far enough”.

#### 2.4.3. Forced-Choice Questions on Alternative Policy Options

A set of five forced-choice questions required respondents to select between alternatives that presented treatment versus preventive health measures and/or individual versus population measures on the basis of which option would make the most difference to improving the community’s health (Question C3, Appendix A). For example, one question compared “taxing processed food with high sugar or fat content” (prevention using regulation) with “subsidising operations for people who are obese” (treatment of individuals). In order to reduce participant burden, only four of the five items were asked of any one participant, with the omitted question selected at random. 

### 2.5. Data Analysis

#### 2.5.1. Scale Creation

A principal component analysis (PCA) was used to create composite measures from the original variables, reflecting respondents’ general positions on government intervention (questions E5a–h, conceptualisations of government intervention) using Stata’s pcf (principal component factor) subcommand with varimax rotation and factor loading of 0.30 threshold for interpretation [28]. PCA is useful for data reduction (to reduce Type 1 error) and capturing aggregate effects [29]. Cronbach’s alpha was calculated over the items to examine the internal reliability of each scale. The analysis showed that there were two components with eigenvalues of 1.0 or more. Items conceptualising government intervention as “nanny statist” constituted the first component. A scale was created by calculating the mean scores for items which had factor loadings >0.30 (E5b, E5c, E5g and E5h, henceforth called the “Nanny State Scale”). Mean scores were calculated to retain the original units for interpretability [30], excluding respondents with missing data for one or more items (n = 150). The remaining questions (E5a, E5d, E5e and E5f) loaded onto a second component but did not demonstrate good internal consistency (Cronbach’s alpha = 0.492), indicating that it was not appropriate for them to comprise a single scale [31]. These questions were therefore used as single items in subsequent analyses to reflect different aspects of non-nanny-statist conceptualisations, namely E5a, paternalistic [26,32]; E5d, utilitarianism [27]; and E5e and E5f, stewardship [25,27]. Previous research has identified these conceptualisations as important alternatives to the nanny state discourse [26,27].

The same procedure was applied to create scales for specific policy measures which reflected commonalities in how respondents supported or opposed different types of policy interventions. Three components were indicated by this analysis: E2d, E2e, E2f, E2h, E2m, E2n and E2p formed a scale which combined questions related to unhealthy foods and alcohol consumption (henceforth called the “Diet Scale”; E2a, E2b and E2p were questions on tobacco regulation (henceforth called the “Tobacco Scale”). The remaining items, E2c, E5j, E2k and E5l, loaded onto a third component; however, the Cronbach’s alpha was poor (0.411) and the items did not converge conceptually upon review. Therefore, these items were not analysed further.

For the forced-choice questions, a scale was created to reflect how much the respondent favoured systemic/regulatory preventive measures over individual-treatment-based measures (henceforth called the “Prevention Preference Score”). As respondents could only choose one of two options, the scale was created by assigning a score of 1 when the respondent selected the systemic/regulatory policy option and a score of 0 for the individual-focused intervention option. The total score was out of 4, as this was the maximum number of questions a respondent could be exposed to. As the items comprising these scales were dichotomous, a series of Kuder–Richardson 20 tests were performed (one for each combination of items) to test for internal consistency. 

For all scales, higher scores indicated greater endorsement of the conceptualisation or support for policy intervention.

#### 2.5.2. Missing Data

Missing data were analysed for the predictor variable(s) (conceptualisations of government intervention) and outcomes. The pattern of missing data was initially checked descriptively and then tested for whether data were missing completely at random (MCAR) using Little’s test [33]. The data were found to be missing at random (MAR). As the analysis entailed linear models, we used full information likelihood models to address the missing data [34].

#### 2.5.3. Descriptive and Regression Analyses

We used external population benchmarks from the year closest to the survey for distributions of age, gender, state and region (capital city/non-capital city), education, country of birth [35] and telephony status (landline only, mobile only, landline and mobile user) [36] to calculate design weights which were the inverse of the probability of a respondent being selected. 

Means and standard deviations for each of the exposure variables and the outcomes were calculated across demographic characteristics (gender, age, metropolitan/non-metropolitan, country of birth (English- vs. non-English-speaking), level of education and socioeconomic status), as these have been shown previously to be associated with opinions on policy options [7]. Statistical analysis was undertaken using Stata version 16.1 [37]. Full information likelihood [34] models were run in Stata using the SEM (structural equation modelling) command with the MLMV (maximum likelihood with missing values) option as recommended [37]. Models included the above demographic variables and main effects for the Nanny State Scale and the four non-nanny-statist conceptualisation items. For the Prevention Preference Score, we included an indicator variable was which coded for the four different subsets of questions comprising the scale to assess for any variation due to scale item composition. Results are presented as the increase in the total score on the scale per unit increase in the scale/item score. R-squared for each model was calculated using the ESTGOF post-estimation command in Stata [37].

## 3. Results

The demographic characteristics of the sample are shown in Table 1. Briefly, over half were female and the majority were aged over 55 years and spoke English at home. Just under two-thirds were living in urban areas, and most were employed or retired (or receiving a pension). In terms of indicators of socioeconomic indicators, just over a third of respondents were living in an area classified as a disadvantaged area and a third received income support, but the sample was fairly evenly split across educational levels. 

### Scale Creation Results

The internal reliability coefficients and measures of central tendency and spread of each scale across demographic characteristics are shown in Table 2. Most of the scales showed moderate reliability, and one showed good reliability (Diet: Cronbach’s alpha = 0.814).

The means and standard deviations of the exposure and outcome variables across demographic characteristics are shown in Table 2 with significance values for the bivariate analyses detailed in Appendix A. In the bivariate analyses (Appendix A), men were more likely to obtain higher scores on the Nanny State Scale and showed lower support for interventions targeting diet and alcohol. Somewhat contradictorily, older respondents (especially those aged over 55) had higher scores on the Nanny State Scale, but were more supportive of policy action for tobacco control and diet than those younger than 35 years. Those respondents with a university degree had significantly lower scores on the Nanny State Scale and higher scores for three of the four non-nanny-statist conceptualisations of government interventions compared with respondents with a high-school education. Those from disadvantaged backgrounds had lower scores on the Nanny State Scale but higher scores on the alternative conceptualisations, reflecting an egalitarian view of government intervention (E5f).

The results of the multiple-variable analyses are shown in Table 3. For the Tobacco Scale, the Nanny State Scale and three of the four alternative conceptualisation items yielded significant results, with the former showing an inverse relationship with support for intervention on tobacco products, and the latter a positive relationship. A similar pattern was observed for both the Diet Scale and the Prevention Preference Score, although only two of the non-nanny-statist items were significant for each (E5d, E5f and E5e, E5f respectively). The change in outcome in all cases was small, no higher than (absolute) 0.15 for any of the main variables of interest. The percentage variance explained increased for all scales with the addition of the Nanny State Scale to the model by between 2.4% (Prevention Preference Score) and 5% (Diet Scale). The addition of the four non-nanny-statist items also increased the R-squared by between 2% (Tobacco Scale) and 7.7% (Diet Scale). The Diet Scale had the highest proportion of variance explained at 17.6% for the full model.

Age was the only consistent correlate among the demographic variables, with those aged over 55 years more supportive of policy action compared with those aged younger than 35 years for all three outcomes, even after taking into account the general position on government intervention. Results for the variable indicating the different subsets of questions comprising the Prevention Preference Score showed that respondents who were not asked question C3d had significantly higher scores on this measure than respondents with other question combinations once adjusted for all other variables (Beta = 0.29 (95%CI 0.09-0.42, *p* = 0.002).

## 4. Discussion

Our analysis of the 2018 AUSPOPS data provides some important insights into the relationship between the general attitudes of Australians towards government intervention for NCD prevention, and their support and preferences for specific preventive interventions. First, measures of the different conceptualisations of government intervention seem to function somewhat independently and are not two ends of the same scale. That is, people could show any combination of positions (for example, high on the Nanny State Scale but also high on any or all of the alternative conceptualisations), potentially endorsing a range of views of government intervention. Our results, therefore, do not support the nanny state notion of intervention as solely interference and total personal responsibility as the only solution [26,38]. In practice, it could mean that in order to promote support for preventive action, communications should reduce invocation of the nanny state argument and instead appeal to alternative conceptualisations such as utilitarianism and stewardship [27]. 

The independence of the nanny-statist view and alternative conceptualisations may explain an earlier analysis of the 2016 AUSPOPS data which showed differences between trends in general community attitudes to government-led prevention and their expressed support for specific interventions [20]. For example, in 2016, older people agreed less with government intervention as a general principle and yet exhibited greater support than their younger counterparts for a wide range of specific interventions such as restrictions on alcohol advertising and lower speed limits in high-pedestrian areas [20]. Our analysis confirms quantitatively what was observed qualitatively in that earlier study, namely that the endorsement of particular interventions is informed by more than an infringement of rights discourse. Further, the poor scaling properties across the four non-nanny-statist conceptualisations demonstrated that scores on one of these items do not predict scores on others. Therefore, although seemingly sharing orthogonality with the nanny-statist view, as a group they do not cohere into a unidimensional non-nanny-statist scale. Advocacy should therefore establish a rationale for support based on a range of factors, as suggested by previous authors, such as intervention effectiveness [39] and stewardship for vulnerable groups [27], as these are more likely to reflect the multiplicity of issues that people take into consideration. 

Despite the complexity of people’s views, we did find that the results were broadly as expected whereby when adjusted for demographic characteristics, higher scores on the Nanny State Scale were associated with lower endorsement of government policy action. The converse was the case for the alternative conceptualisations of government intervention—higher scores for these items were correlated with greater endorsement of action. The alternative conceptualisation capturing the stewardship model of government intervention [25,27], “Limiting the advertising and sale of unhealthy products makes it easier for people to make healthy choices” (E5f), was correlated with all three scales (Tobacco, Diet and Prevention Preference) when adjusted for all other variables. As a conceptualisation which takes an egalitarian world view [16], it supports system-level responses, such as legislation, to assist citizens to act in health-promoting ways. As noted in the introduction, upstream interventions are the most effective for producing change and preserve and support individual agency. Our analysis therefore supports the link between a stewardship conceptualisation and desire for policy action related to tobacco and diet. 

Between different target behaviours and interventions, the Tobacco Scale had the highest number of significant relationships with the different conceptualisations, with only the shared responsibility item not reaching statistical significance. A number of factors may explain these results. Tobacco control has achieved health-promotion success as exemplified by declining smoking rates and making smoking a non-normative behaviour in the community [40]. Further, tobacco control has been supported by regulatory measures such as tax increases, plain packaging and environmental restrictions on where people can smoke [41]. The now well-accepted direct impacts of smoking and the potential to affect the health of those other than the smoker (i.e., through passive smoking) may mean that the group-minded approaches encapsulated in the alternative conceptualisations closely connect with the restrictive measures comprising the Tobacco Scale. The Diet Scale, however, showed better prediction of community perceptions overall, suggesting that demographic characteristics and general attitudes towards government intervention to some extent account for positions on this set of measures. Diet is a more complex behaviour than smoking and, in this scale, includes alcohol consumption. Moreover, the current prevalence of tobacco use in Australia is low compared with alcohol consumption and unhealthy eating [42], and in general people are more supportive of regulation which does not affect them directly [43]. There is also evidence that even smokers are supportive of tobacco control policy [44]. Therefore, our results may show the combined effect of these contextual factors as support for tobacco control irrespective of a person’s general disposition regarding government intervention. Previous research has shown lower support in the community for actions restricting individuals’ alcohol and food consumption, but good support for more upstream interventions such as imposing bans and limiting advertising, as reflected in the items comprising this scale. A preference for regulation of commercial interests over individuals’ in relation to diet would also explain why the items reflecting utilitarian (E5d) and stewardship (E5f) conceptualisations were significant, but the paternalism item (E5a) was not. Our findings support upstream targeting of factors such as food production, marketing and corporate or commercial responsibility, and avoiding industry-preferred arguments around personal responsibility and individual choice [45,46]. Importantly, there is also evidence that these types of interventions are also more effective at promoting healthy diets [47].

The demographic factor which emerged as most strongly related to outcomes was older age. Those aged 55 and older demonstrated greater support for policy action on tobacco and diet, and greater preference for system-level rather than individual-level interventions. The relationship between age and policy support in the AUSPOPS study has been discussed in detail elsewhere [20,22], but it is interesting to note that this effect persisted even after taking into account the general disposition of the respondents to intervention. It could be that because the interventions asked about in this survey were mostly aimed at system-level change or targeted industry behaviour, the conservatism demonstrated by older people in terms of policy change related to individual-behaviour-focused social issues (e.g., marriage equality [48]) was not evident. Alternatively, older people may have more chronic conditions themselves, and take the need to act on “wicked problems” more seriously. Future research could explore further the means by which groups with higher opposition to intervention in general but support for specific interventions reconcile these positions. There were only two other significant demographic associations: those born in non-English-speaking countries responded more favourably about tobacco interventions and women were more supportive than men for policy action related to diet, consistent with previous research [49,50]. Interestingly, neither of the indicators of socioeconomic status (area-level SES and education) were associated with the outcomes, contrary to previous reviews showing significant relationships [7,50]. However, the difference may be attributable to differences in the measures used for SES (we used education and area-level SES and others have used income).

Effect sizes in these associations were generally small and hence not fully explanatory. It may be, as with the work conducted in the area of cultural cognition [16], that the measures themselves need further refinement to better represent latent constructs around different conceptualisations. Although the question items were grounded in formative evaluation, there may be a need for further testing with community members to ensure the concepts are being captured as intended. From an operationalisation perspective, respondents may additionally need to rank the conceptualisation which most strongly represents their general position rather than rating them independently of one another, to better capture relative strength. Conceptually, however, it may be as it appears—that people do not necessarily align themselves in any kind of simplistic way when making judgements about specific interventions. Certain aspects such as the type of intervention, the target risk behaviour and the risk of collateral harm may interact unpredictably with the different conceptualisations, and particularly with the alternative conceptualisations. Similar conceptualisations have arisen in research investigating COVID-19 interventions. For example, in one study from the United Kingdom, decisions about vaccination were informed by notions of collective responsibility (utilitarianism) and perceptions of the authenticity of government intervention [51]. A similar finding in relation to social-distancing compliance for COVID-19 was found in the United States, where compliance was greater among group-oriented (vs. self-focused) individuals [52]. Our analysis here is more hypothesis-generating than leading to a unifying theory. Further qualitative work is needed to better understand the mechanisms underlying these relationships in NCD prevention and other health policy areas.

### Strengths and Limitations

To our knowledge, this is the first study to explore the relationship between general positions on NCD prevention and level of endorsement of specific interventions. The sample was representative of the Australian population thanks to rigorous sampling and weighting procedures, and analyses were adjusted for demographic factors which have been previously associated with public opinion. While our questions chosen to reflect alternative conceptualisations of government intervention were grounded in qualitative formative analyses, there may be other measures which may be more suited to tapping into an underlying broad concept around non-nanny-statist positioning. Measures of support were mostly sought for interventions at the system level rather than those which were more intrusive into individual behaviour; therefore, the patterning of results might differ with different targeting. Although results were statistically significant, the small effect sizes and low to moderate amount of explained variation suggests that there is still much unexplained variance in our outcome variables. Finally, as an exploratory study with limited measures of both the outcome and predictor concepts, developing a unified theory of the relationship between general and specific positions on government intervention was beyond the scope of this analysis. However, we linked the results to previous theoretical and empirical literature in order to interpret the findings beyond description alone.

## 5. Conclusions

Our analysis showed that general positions on government intervention are correlated, but not specific to, support for specific interventions related to tobacco control and diet, and a preference for system-level over individual behavioural measures. Advocacy communications related to government intervention should be conscious of the conceptualisation of government in their messaging. Our results suggest that using utilitarian [27] and/or stewardship conceptualisations [25,27] may have broad appeal and avoid inadvertently invoking a rights narrative. Future research could use different methods, such as relative rankings, to clarify the independence of these conceptualisations.

## Figures and Tables

**Table 1 healthcare-11-01246-t001:** Demographic profile of 2018 AUSPOPS sample n = 2601 (unweighted).

Characteristic	No.	%
Female	1364	52.4%
Age		
18–<35 years	429	16.5%
35–<55 years	738	28.4%
55+ years	1432	55.1%
Metropolitan (vs. regional)	1603	62.4%
Country of birth: English-speaking ^a^	2183	84.0%
English spoken at home	2266	87.1%
Aboriginal or Torres Strait Islander	54	2.1%
Employment status		
Employed	1343	51.8%
Unemployed	72	2.8%
Retired/pension	957	36.9%
Student/home duties/other	222	8.6%
Highest level of education		
High school	832	32.8%
Post-secondary	822	32.4%
University degree	883	34.8%
Disadvantaged ^b^	904	35.2%

^a^ Australia, New Zealand, United Kingdom (England, Scotland, Wales, Northern Ireland), USA, Canada. ^b^ SEIFA Index of Relative Disadvantage quintiles 1–2.

**Table 2 healthcare-11-01246-t002:** Internal reliability, means and standard deviations of exposure and outcome variables across demographic variables (see Appendix A for significance values of bivariate comparisons).

Measure								
	Nanny State Scale	Non-Nanny-State Conceptualisations				Tobacco Scale	Diet Scale	Prevention Preference Score
		E5a	E5d	E5e	E5f			
Range	1–5	1–5				1–3	1–3	0–4
Internal reliability *,†	0.706	NA	NA	NA	NA	0.633	0.814	0.677–0.759 §
Characteristic	Mean (SD)	Mean (SD)	Mean (SD)	Mean (SD)	Mean (SD)	Mean (SD)	Mean (SD)	Mean (SD)
All	2.86 (1.00)	3.93 (1.18)	2.70 (1.36)	4.43 (0.90)	3.89 (1.26)	2.25 (0.48)	2.37 (0.47)	2.76 (1.04)
Male (ref.)	2.97 (0.98)	3.87 (1.19)	2.66 (1.36)	4.40 (0.91)	3.84 (1.28)	2.23 (0.48)	2.32 (0.48)	2.75 (1.05)
Female	2.76 (1.01)	3.99 (1.17)	2.73 (1.35)	4.47 (0.88)	3.95 (1.22)	2.27 (0.48)	2.42 (0.46)	2.77 (1.02)
Age	<0.001	0.650	0.992	0.022	0.998	<0.001	<0.001	0.142
18–<35 years (ref.)	2.75 (0.67)	3.98 (0.80)	2.70 (0.96)	4.45 (0.6)	3.89 (0.90)	2.20 (0.30)	2.24 (0.33)	2.71 (0.73)
35–<55 years	2.77 (0.94)	3.91 (1.10)	2.69 (1.25)	4.49 (0.79)	3.89 (1.17)	2.20 (0.44)	2.39 (0.44)	2.75 (0.97)
55+ years	3.06 (1.31)	3.91 (1.55)	2.70 (1.75)	4.36 (1.22)	3.89 (1.57)	2.35 (0.66)	2.47 (0.57)	2.83 (1.33)
Metropolitan (ref.)	2.78 (0.95)	3.97 (1.11)	2.72 (1.32)	4.47 (0.85)	3.94 (1.20)	2.26 (0.47)	2.37 (0.45)	2.77 (1.02)
Regional	3.02 (1.08)	3.85 (1.31)	2.66 (1.44)	4.36 (0.98)	3.82 (1.34)	2.23 (0.51)	2.36 (0.50)	2.75 (1.07)
Country of birth English-speaking ^a^ (ref.)	2.89 (1.07)	3.90 (1.24)	2.65 (1.40)	4.45 (0.92)	3.84 (1.34)	2.23 (0.51)	2.36 (0.50)	2.73 (1.11)
Country of birth not English-speaking ^a^	2.77 (0.77)	4.04 (0.95)	2.83 (1.17)	4.39 (0.78)	4.05 (0.97)	2.32 (0.38)	2.4 (0.37)	2.87 (0.81)
Highest level of education	<0.001	<0.001	<0.007	0.001	0.169	0.444	0.049	0.001
High school (ref.)	3.05 (0.98)	4.01 (1.14)	2.68 (1.32)	4.28 (0.98)	3.86 (1.26)	2.27 (0.49)	2.35 (0.46)	2.66 (1.02)
Post-secondary	2.97 (0.89)	3.76 (1.14)	2.62 (1.25)	4.48 (0.76)	3.87 (1.16)	2.23 (0.44)	2.37 (0.44)	2.75 (0.96)
University degree	2.48 (1.05)	4.10 (1.18)	2.87 (1.52)	4.59 (0.88)	3.99 (1.34)	2.26 (0.50)	2.42 (0.53)	2.91 (1.13)
Not disadvantaged (ref.)	3.08 (1.08)	3.85 (1.35)	2.63 (1.46)	4.37 (0.98)	3.71 (1.45)	2.22 (0.54)	2.34 (0.53)	2.70 (1.15)
Disadvantaged ^b^	2.77 (0.95)	3.96 (1.11)	2.73 (1.31)	4.46 (0.86)	3.98 (1.15)	2.26 (0.45)	2.39 (0.45)	2.79 (0.99)

† All items and scales range 1–5, except Prevention Preference Score which is 1–4. * Cronbach’s alpha for all scales except Prevention Preference Score which was tested using the Kuder–Richardson (KR20) test. § The KR20 was undertaken for each combination of 4 of the 5 constitutive questions in this scale and the minimum to the maximum reliability coefficients are shown. ^a^ Australia, New Zealand, United Kingdom (England, Scotland, Wales, Northern Ireland), USA, Canada). ^b^ SEIFA Index of Relative Disadvantage quintiles 1–2.

**Table 3 healthcare-11-01246-t003:** Beta coefficients for regression analyses of four outcome scales for Nanny State Scale and non-nanny-statist questions (E5a, E5d, E5e, E5f) adjusted for demographic variables.

	Scale		
	Tobacco Scale	Diet Scale	Prevention Preferences Score
R-squared			
Model 1 ^a^	3.3%	5.9%	2.4%
Model 2 ^b^	6.7%	10.9%	4.8%
Model 3 ^c^	8.7%	17.6%	6.9%
Characteristic (ref. cat.)			
*Conceptualisations*			
Nanny State Scale	−0.07 (−0.10, −0.04)	−0.08 (−0.11, −0.06)	−0.15 (−0.22, −0.09)
E5a	0.03 (<0.01, 0.05)	<−0.01 (−0.02, 0.02)	−0.03 (−0.09, 0.02)
E5d	0.02 (0.01, 0.04)	0.03 (0.02, 0.05)	−0.01 (−0.05, 0.04)
E5e	<0.01 (−0.03, 0.03)	0.02 (<−0.01, 0.05)	0.09 (0.02, 0.15)
E5f	0.03 (0.01, 0.05)	0.08 (0.06, 0.11)	0.10 (0.05, 0.14)
*Demographic covariates*			
Female (male)	0.01 (−0.03, 0.06)	0.06 (0.02, 0.11)	−0.02 (−0.13, 0.09)
Age (18–<35 years)			
35–<55 years	0.02 (−0.04, 0.08)	0.15 (0.09, 0.21)	0.03 (−0.12, 0.17)
55+ years	0.17 (0.12, 0.23)	0.26 (0.20, 0.31)	0.19 (0.06, 0.32)
Regional (metropolitan)	0.02 (−0.03, 0.08)	0.02 (−0.03, 0.08)	0.07 (−0.05, 0.20)
Country of birth not English-speaking ^d^ (English-speaking)	0.10 (0.04, 0.17)	0.03 (−0.03, 0.09)	0.13 (−0.01, 0.26)
Highest level of education (high school)			
Post-secondary	−0.01 (−0.07, 0.05)	0.01 (−0.04, 0.07)	0.06 (−0.07, 0.2)
University degree	−0.05 (−0.11, 0.01)	0.02 (−0.04, 0.08)	0.12 (−0.02, 0.26)
Disadvantaged ^e^ (not disadvantaged)	−0.02 (−0.08, 0.03)	−0.01 (−0.06, 0.05)	−0.02 (−0.15, 0.12)

^a^ Model 1—includes demographic variables only (plus question subset indicator for Prevention Preference Score) ^b^ Model 2—includes demographic variables and Nanny State Scale (plus question subset indicator for Prevention Preference Score) ^c^ Model 3—includes demographic variables, Nanny State Scale and non-nanny-statist conceptualisations (plus question subset indicator for Prevention Preference Score) ^d^ Australia, New Zealand, United Kingdom (England, Scotland, Wales, Northern Ireland), USA, Canada) ^e^ SEIFA Index of Relative Disadvantage quintiles 1–2.

## Data Availability

The datasets generated and/or analysed during the current study are not publicly available as such access would be a breach of the original ethics approval for the study, but are available from the corresponding author on reasonable request.

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
