# Peer review of "Are Perceptions of Government Intervention Related to Support for Prevention? An Australian Survey Study"

_healthcare, 2023, doi:10.3390/healthcare11091246_

Round 1

Reviewer 1 Report

Comments to the authors:

The reviewer examined the submitted paper titled “Are perceptions of government intervention related to support for prevention? An Australian survey study,” with considerable interest. This study clarified that the image of the Nanny state held by people was negatively associated with support for health intervention policies implemented by governments. Additionally, the study indicated that cohorts have significant effects on support for health intervention policies; the elderly cohorts tend to support such intervention policies more than younger cohorts. The reviewer believes that these findings have significant implications for social researchers and policymakers. Therefore, this study might contribute to the existing literature.

However, as the reviewer has some concerns regarding the submitted paper, he hesitated to recommend it for publication. The authors followed a strictly scientific procedure to examine the effect of perceptions of government intervention on support for health intervention policies. The reviewer believes that the submitted paper could be highly evaluated on the point. However, the English language used in the submitted paper was suffering and not readable, at least for a non-native English speaker such as the reviewer. The reviewer expects that the authors will examine the English language of the submitted paper to improve its readability. Additionally, the reviewer has concerns about the theoretical explanations and methods adopted by the authors. The reviewer expects that the authors will reexamine their paper according to his suggestions and comments.

First, although the authors presented interesting findings of their study, the reviewer could not find a substantial theory to explain them. Similarly, the reviewer did not find any specific hypotheses derived from the authors’ theoretical explanations in the submitted manuscript. Consequently, the submitted paper appears to be a descriptive study that lacks theory. The reviewer states that the authors should examine their findings theoretically and, by doing so, argue how they apply to other intervention policies related to people’s health. If the authors had theoretically examined their findings, they would have more significant implications for health policy studies. For instance, while the meaning of the Nanny state was intuitively understandable, the meanings of the other variables of conceptualization (E5a, E5d, E5e, and E5f) were unclear. Consequently, although some of the conceptualization variables had statistically significant effects on health intervention policies, these findings do not contribute to the literature. However, if the authors theoretically examined the meanings of each variable in E5a, E5d, E5e, and E5f, the reviewer confirms that these findings might contribute to the literature.

Next, the authors implemented a principal components analysis (PCA) and estimated linear regression models predicting the evaluations of health intervention policies using the factors extracted from the PCA. However, the authors stated that they used the command of structural equation modeling to estimate the regression models. The reviewer wonders why the authors did not use latent factors extracted from the exposure variables and specific policy measures when implementing structural equation modeling. They could simultaneously and efficiently estimate the coefficients of variables and latent factors using structural equation modeling. If the authors have reasons to avoid using the latent variables, the reviewers believe that they should explain this in their paper. If they have no reason to avoid using it, they should implement models that include latent variables.

Related to this, the authors did not present a path diagram to denote the relations with variables in their conceptual (or theoretical) model. If they had prepared a path diagram, it could have improved the readability and theoretical clarity of the submitted paper.

Finally, the reviewer believes that the authors should discuss the implications of their findings in the field of health policy studies more clearly and generally. For instance, the reviewer believes that the findings of the authors’ study could be applied to the problem of COVID-19 vaccination hesitancy, which has emerged during the COVID-19 pandemic.

The reviewer is grateful for the opportunity to review this paper. He sincerely expects that his comments will contribute to improving this paper.

Reviewer 2 Report

The paper is a relevant contribution to the analysis of opinions on government intervention. As noted, it is the first study of its kind, so it is possible that the methodology could be replicated in other contexts.

It is suggested that in the conclusion it be noted whether, apart from the different relative ranking methods, it would be possible to incorporate other specific interventions or alternative conceptualizations in future studies.  

I suggest revising this part of the last paragraph ("Future research" is repeated): "Future research could Future research should use different methods, such as relative rankings, to clarify the independence of these conceptualizations."

Reviewer 3 Report

The manuscript deals with a study in Australia on the conceptualization of government interventions and support of different policy options for tobacco and diet.

The introduction part restricts the discussions to issues of NCD prevention interventions. Reference is made to policies on tobacco, diet, alcohol, physical actrivity and obesity. It would be interesting for comparison to make reference also to traffic safety policies. The empirical literature is biased to Australian experience that, howerer, is much studied. Important notion is how peer review lterature is so much dominated by individual behaviour measures.

The material is based on a commercial sample provider using Random Digit Dialling and with set specific procedures. In the limitations part  of the manuscript the problems of the procedures and the representativeness of the material should be furter discussed, although the analyses on missing data seem to be good. The possible problems with the material unlikely influence the main results.

The measures are set of questions, either previously used or developed from qualitative focus group experience. Qualitative studies on this kind of issues are clearly helpful and add to the information from quantitative studies.

The principal component analyses are useful in giving the used "Nanny State Scale", "Diet Scale", "Tobacco Scale" and "Prevention Preference Scale".

Table 2 results are interesting, showing e.g. that men were more likely to have higher scores on the nanny scale and lower support for diet and alcohol interventions. An unexpected finding was that older persons had higher score on nanny state, but were more supportative of tobacco and diet policies. University degree and coming from disadvantaged backgrounds had also an influence.

The important finding was that the nanny scaled had inverse relationship with Tobacco, Diet and Prevention Preference scores, but the effect sizes in all cases were surprisingly small.

I find the discussion part very good. It has important conclusions: "Measures of different conceptualizations of government iterventions seem to function somewhat independently and are not the ends of the same scale."! The implcations of this finding for health promotion and policies should be further discussed.

The discussion comments interestingly and plausibly why tobacco scale had the most significant relationship with the different conceptualizations. Althogh page 9 has good comments, we should discuss more why tobacco policy in the Western world has good public support, while diet and alcohol policies face strong opposition - even if they all have big commercial actors.

The main sentence in the conclusion part is: "Advocacy communications for government iterventions should be concious of conceptualization of goverment which is implied to ensure a framing which has broadest appeal and avoid inadvertently invoking a rights narrative." This complicated message should be given in more plain English, at least for non-native English speakers!

Round 2

Reviewer 1 Report

The reviewer examined the revised paper titled “Are perceptions of government intervention related to support for prevention? An Australian survey study,” with considerable interest. Unfortunately, the reviewer believes that the revised paper does not fully reflect his comments. On the other hand, the reviewer confirmed that the authors minimally responded to his comments. While the reviewer felt that the revised manuscript could be improved further, he judged that it met the criteria for publication in the journal. Therefore, the reviewer would like to recommend the revised manuscript for publication.

The reviewer is grateful for the opportunity to review this paper. He sincerely hopes the success of the authors’ study.